# Isolation, Characterization and Pharmacological Investigations of a New Phenolic Compound along with Four Others Firstly Reported Phytochemicals from *Glycosmis cyanocarpa* (Blume) Spreng

**DOI:** 10.3390/molecules27185972

**Published:** 2022-09-14

**Authors:** Sania Ashrafi, Safaet Alam, Nazim Uddin Emon, Monira Ahsan

**Affiliations:** 1Department of Pharmaceutical Chemistry, University of Dhaka, Dhaka 1000, Bangladesh; 2Drugs and Toxins Research Division, BCSIR Laboratories Rajshahi, Bangladesh Council of Scientific and Industrial Research, Rajshahi 6206, Bangladesh; 3Department of Pharmacy, Faculty of Science and Engineering, International Islamic University Chittagong, Chittagong 4318, Bangladesh

**Keywords:** *Glycosmis cyanocarpa*, phytochemical, NMR, antioxidant, cytotoxicity, antibacterial, polyphenol, alkaloid

## Abstract

Plants are serving the mankind with important bioactive phytochemicals from the very ancient ages to develop novel therapeutics against different disease states. *Glycosmis cyanocarpa* (Blume) Spreng is a plant from the Rutaceae family and a very less explored species from the Glycosmis genus. Thus, this present study was intended to present the chemical and biological investigation of *Glycosmis cyanocarpa* (Blume) Spreng. The chemical investigation resulted in the isolation of one new phenolic compound to the best of our knowledge which is (4-(3-hydroxy-2-methylpropyl)-2-methoxyphenol) (**1**) along with four known compounds that are isolated for the first time from this species- 3-methyl-1*H*-indole (**2**), Tri-*trans*poly-*cis* prenol-12 (**3**), Stigmasterol (**4**) and β-sitosterol (**5**). Their chemical structures were elucidated based on extensive spectroscopic methods, including 1D and 2D NMR, and comparison with the available literature data. Isolated phytochemicals were further investigated to unveil their antioxidant properties with IC_50_ values (ranged from 9.97–75.48 µg/mL), cytotoxicity with LC_50_ values (ranged from 1.02–1.92 µg/mL), and antibacterial properties against some selected Gram (+) ve and Gram (−) ve bacteria. Among the compounds, 3-methyl-1*H*-indole (**2**) was found to be the most active against *Staphylococcus aureus*. Moreover, the phenolic compound (**1**) and the alkaloid (**2**) revealed the highest antioxidant (9.97 µg/mL) and cytotoxic activities (1.02 µg/mL), respectively. Thus, the isolation of these bioactive phytochemicals from the plant revealed a new perception in the study arena of drug discovery and the findings may ease the development and discovery of novel therapeutics. Further investigations are still recommended to understand their exact molecular mechanism and toxicological impact.

## 1. Introduction

Since the dawn of human civilization, various diseases have been treated with medicines and compounds derived from plants [1,2,3,4]. Since the beginning of time, all cultures have made considerable use of plants to enhance health and treat a variety of maladies. According to the World Health Organization (WHO), traditional medicine is the primary source of health care for 80% of people worldwide [5,6]. Numerous secondary metabolites generated from plants have been identified to have beneficial effects on the liver, including hepatoprotection, cytotoxicity, antidepressants, antioxidants and antidepressants [7].

Secondary metabolites are the wide range of chemical components produced by plants, but these secondary metabolites have no direct role in plant development and reproduction [8,9,10,11]. Plant secondary metabolites serve a number of purposes, including those related to defence response signalling [12], innate immunity [13] and plant growth and development processes [14]. These chemicals have pharmacological effects on the human body [15]. This is due to their remarkable biological activities, flavonoids, cyclitol [16], alkaloids, phenolic compounds and tannins [17]. Therefore, the separation and purification of secondary metabolites from plants are becoming increasingly popular and numerous separation techniques are extensively used for extraction, separation and purification of secondary metabolites [16].

*Glycosmis cyanocarpa* (Blume) Spreng. belongs to the family Rutaceae. This is a small tree or shrub. Flowers are constantly 4-merous with 8 stamens and 4-locular ovaries. Fruits are longer than wide, oblong, ellipsoid or ovoid. The plant is distributed over India, Nepal, South Tibet, Bangladesh, Burma, Sri Lanka, Thailand, Malaysia, Philippines and Indonesia (western) [18]. The plant has produced numerous phytochemicals including quinolone alkaloid, cinnamides, nor-diterpenes and sulfur-containing amides that possess antifungal properties [19,20,21,22,23].

Reactive oxygen species (ROS) are formed when there is an imbalance between the defense provided by the antioxidant system and the production of ROS and it leads to the oxidation of lipids, blood vessel walls, proteins, carbohydrates, DNA, etc. [24,25]. In terms of severity, after cardiovascular disorders, cancer is one of the main causes of death worldwide. Around the world, 182 out of every 100,000 people battle cancer annually, and 102 people lose their lives to the disease. According to a World Health Organization (WHO) report, there are 14 million people globally who are living with cancer, and there are 8 million fatal cases [26]. A total of 19.3 million new cases of cancer and approximately 10 million cancer deaths globally were reported in statistics provided by the World Health Organization International Agency for Cancer Research [27]. According to reports, 13 percent of deaths in Bangladesh are caused by cancer [28]. Antibiotic resistance, a pressing problem in public health today, arises from the haphazard use of commercially available antibiotic medications. Nearly 50,000 people die from Methicillin-resistant *Staphylococcus aureus* (MRSA) each year in the United States and Europe alone, with many more dying from it in other regions [29]. In 2018, in contrast to the 66 nations, territories and areas where infectious illnesses were recorded, 2,164,568 cases of antibiotic resistance were reported in the Global Antimicrobial Resistance and Use Surveillance System (GLASS) Report, published by the WHO in 2020 [30]. As a result, we require antibiotics with increased dosages and fewer side effects. Herbal medicine is a highly practical solution in this case [31]. However, the use of standard treatment methods is restricted by side effects, contraindications, non-selectivity of chemotherapy medications, antibiotic resistance, hazardous responses and high cost. As a result, it is hoped that natural products will be more accessible and have fewer side effects [32,33,34,35].

In this study, we investigated *G. cyanocarpa* and identified five chemicals from the plant’s methanol extract. In addition, the antioxidant, cytotoxic and antibacterial properties of isolated phytochemicals.

## 2. Results

### 2.1. Isolated Phytochemicals from G. cyanocarpa

Five compounds (Figure 1) were isolated from the crude methanol extract of the stem and leaf of *G. cyanocarpa* by following repeated chromatographic separations. The structures of the isolated compounds were elucidated as (4-(3-hydroxy-2-methylpropyl)-2-methoxyphenol) (**1**), 3-methyl-1*H*-indole (**2**) [36], Tri-*trans*poly-*cis* prenol-12 (**3**) [37], Stigmasterol (**4**) [38] and β-sitosterol (**5**) [38] by analyzing the NMR spectral data and comparing those data with published values. 

(4-(3-hydroxy-2-methylpropyl)-2-methoxyphenol) (**1**): Yellow mass and soluble in methanol and chloroform; ^1^H NMR (400 MHz, CDCl_3_): δ6.53 (1H br s, H-2), 6.82 (1H d, *J* = 8.0 Hz, H-5), 6.60 (1H d, *J* = 8.0 Hz, H-6), 2.57 (2H br t, *J* = 6.8 Hz, H-7), 2.55 (1H m, H-8), 3.93 (2H m, H-9), 3.85 (3H s, OCH_3_-3), 0.86 (3H m, CH_3_-8). ^13^C NMR (100 MHz, CDCl_3_): δ132.32 (C-1), 111.05 (C-2), 146.40 (C-3), 143.90 (C-4), 114.09 (C-5), 121.32 (C-6), 46.47 (C-7), 39.23 (C-8), 73.30 (C-9), 55.79 (OCH_3_-3), 14.14 (CH_3_-8) (Appendix A).

3-methyl-1*H*-Indole (**2**): Yellow mass and soluble in methanol and chloroform; ^1^H NMR (400 MHz, CDCl_3_): δ7.04 (1H d, *J* = 2.0 Hz, H-2), 7.66 (1H d, *J* = 7.8 Hz, H-4), 7.13 (1H dd, *J* = 7.8, 7.2 Hz, H-5), 7.2 (1H dd, *J* = 7.8, 7.2 Hz, H-6), 7.36 (1H d, *J* = 8.0 Hz, H-7), 7.99 (1H d, *J* = 2.0 Hz, NH), 2.36 (3H s, CH_3_-3). ^13^C NMR (100 MHz, CDCl_3_): δ121.96 (C-2), 100.61 (C-3), 118.73 (C-4), 119.56 (C-5), 121.96 (C-6), 110.20 (C-7), 139.50 (C-8), 136.30 (C-9), 13.06 (CH_3_-3) (Appendix A).

Tri-*trans*poly-*cis* prenol- 12 (**3**): Colorless mass and soluble in methanol and chloroform; ^1^H NMR (400 MHz, CDCl_3_): δ4.09 (2H d, *J* = 7.2 Hz, H-1), 5.44 (1H t, *J* = 7.2 Hz, H-2), 1.74 (3H s, H-3), 2.03 (44H m, H-4,5,8,9,12,13, 16,17,20,21,24,25, 28,29, 32,33,36,37,40,41,44,45), 5.12 (11H br s, H-6,10,14,18,22,26,30,34,38,42,46), 1.60 (12H s, CH_3_-trans 35,39,43 and ω- CH_3_- trans), 1.68 (24H s, CH_3_-cis 7,11,15,19,23,27,31 and ω- CH_3_-cis) (Appendix A).

Stigmasterol (**4**): White crystal and soluble in methanol and chloroform; ^1^H NMR (400 MHz, CDCl_3_): δ3.52 (1H m, H-3), 5.35 (1H d, *J* = 4.2 Hz, H-6), 0.70 (3H s, H-18), 1.01 (3H s, H-19), 1.02 (3H d, *J* = 7.6 Hz, H-21), 5.15 (1H dd, *J* = 15.1, 8.6 Hz, H-22), 5.01 (1H dd, *J* = 15.1, 8.6 Hz, H-23), 0.80 (3H d, *J* = 6.8 Hz, H-26), 0.85 (3H d, *J* = 6.1 Hz, H-27), 0.80 (3H t, *J* = 7.0 Hz, H-29) (Appendix A).

β-sitosterol (**5**): White crystal and soluble in methanol and chloroform; ^1^H NMR (400 MHz, CDCl3): δ3.55 (1H m, H-3), 5.37 (1H d, *J* = 4.8 Hz, H-6), 0.71 (3H s, H-18), 1.07 (3H s, H-19), 0.94 (3H d, *J* = 7.6 Hz, H-21), 0.82 (3H d, *J* = 6.5 Hz, H-26), 0.84 (3H d, *J* = 6.5 Hz, H-27), 0.86 (3H t, *J* = 7.6 Hz, H-29) (Appendix A).

The ^1^H NMR spectrum (400 MHz, CDCl_3_) of compound **1** showed three aromatic protons at δ 6.53 br s, 6.82 1H d and 6.60 1H d (*J* = 8.0 Hz each) indicating a trisubstituted benzene ring with ABX system. A three-proton singlet at δ 3.85 was assigned to a methoxy group at C-3 position. The proton directly adjacent to aromatic ring resonated at δ 2.57 (C-7). The chemical shift values of the protons of the compound are very similar to those of 2-methoxy-4-propyl phenol [39] except that the proton at C-8 position showed resonance at δ 1.59 2H dd (*J* = 7.2, 7.7 Hz) instead of δ 2.55 1H m. The spectrum also revealed the presence of methyl group at δ 0.86 3H m which must be assigned to C-8. The ^13^C NMR spectrum indicated 11 carbons including a methyl carbon at δ 14.14 and a methoxy carbon at δ 55.79. Position of methyl group was confirmed at C-8 by an HMBC experiment where they showed ^2^*J* correlation to C-8. On basis of these data and comparing with the previously published literature, Compound **1** was identified as 4-(3-hydroxy-2-methylpropyl)-2-methoxyphenol which is a new natural compound.

The indole and 2-quinolone alkaloids are commonly found in species of the Rutaceae family [40,41]. The ^1^H NMR spectrum (400 MHz, CDCl_3_) of compound **2** displays a doublet of doublet at δ 7.20 (H-6, *J* = 7.8 Hz, 7.2 Hz) and a doublet at δ 7.04 (*J* = 2.0 Hz). Thus, the spectrum revealed a disubstituted indole ring with four aromatic protons comprising an ABCD ring system with expected coupling constants. The methyl group at C-3 appeared at δ 2.36 (3H s). The H-2 proton appeared as a doublet instead of a singlet as it coupled to NH proton which also appeared in the ^1^H NMR spectrum as a doublet with a small coupling constant of 2.0 Hz. This was supported by the COSY correlation as the H-2 proton at δ 7.04 shows a cross peak with NH proton at δ 7.99 (Figure 2). The ^13^C NMR and the HSQC spectra helped to assign all the protons and carbons of this indole. Thus, the compound was identified as 3-methyl-1*H*-Indole. The structure was further confirmed by comparison of its ^1^H NMR spectral data with those published [36].

Compound **3** was identified as polyprenol-12. It is a long chain isoprenoid alcohol containing twelve isoprene units which consists of a dimethylallyl terminal unit (ω-terminal), three *trans*-isoprene units, seven *cis*-isoprene units and a hydroxylated *cis*-isoprene unit (α-terminal). The occurrence of long chain polyprenols (polyprenol-13 and polyprenol-10) in Rutaceae was first reported in 1992 from the plant *Esenbeckia nesiotica* [42]. The ^1^H NMR spectrum (400 MHz, CDCl_3_) displayed the olefinic proton of the α-terminal isoprenol unit at δ 5.44 (1H t, *J* = 7.2 Hz) and the methylene protons at δ 4.09 (2H d, *J* = 7.2 Hz). A signal resonating at δ 1.74 (3H s) could be assigned to the *cis*-methyl of the α-terminal isoprene unit [37]. The signal at δ 1.60 integrated for 12 protons are assigned to three internal *trans*-methyl and a ω-terminal *trans*-methyl group. The spectrum further showed the presence of eight *cis*-methyls resonating at δ 1.68 (24H s) of which seven are internal *cis*-methyl and the rest is ω-terminal *cis*-methyl. In addition, the spectrum showed 22 methylene groups at δ 2.03 (44H m) of the 11 isoprene units. On this basis, the compound was identified as tri-*trans*poly-*cis* prenol-12. This is the first report of occurrence of polyprenol in the genus *Glycosmis*.

The ^1^H NMR spectrum (400 MHz, CDCl_3_) of compound **4** revealed a one proton multiplet at δ 3.52, the position and multiplicity of which was indicative of H-3 of the steroidal nucleus. The typical signal for the olefinic H-6 of the steroidal skeleton was evident item a multiplet at δ 5.35 integrating one proton. The olefinic protons (H-22 and H-23) appeared as characteristic downfield signals at δ 5.01 and δ 5.15, respectively, in the ^1^HNMR spectrum. Each of the signal was observed as doublet of doublets (*J* = 15.0 Hz, 6.5 Hz) which indicated couplings with the neighboring olefinic and methine protons. The spectrum further revealed signals at δ 0.70 and δ 1.01 (3H each) assignable to two tertiary methyl groups at C-18 and C-19, respectively. The ^1^H NMR spectrum also showed two doublets centered at δ 0.80 and δ 0.85 which could be attributed to the methyl groups at C-25. The doublet at δ1.02 (*J* = 6.4 Hz) was demonstrative of a methyl group at C-20. On the other hand, the triplet (*J* = 6.5 Hz) of three-proton intensity at δ 0.80 could be assigned to the primary methyl group attached to C-28. The above spectral features are in close agreement to those observed for stigmasterol [38]. On this basis, the identity of GCL-156 was confirmed as stigmasterol. Stigmasterol is described for the first time from *Glycosmis cyanocarpa*.

The ^1^H NMR spectrum (400 MHz, CDCl_3_) of compound **5** displayed a one proton multiplet at δ 3.55; the position and multiplicity of which was indicative to H-3 of a steroid nucleus. The typical olefinic H-6 of the steroidal skeleton was evident as a doublet at δ 5.37 that integrated for one proton. The spectrum also revealed signals at δ 0.71 and δ 1.07 (3H each) assignable to two tertiary methyl groups at C-13 (H-18) and C-10 (H-19), respectively. The ^1^H NMR spectrum showed two doublets (*J* = 6.5 Hz) centered at δ 0.82 and δ 0.84 which could be attributed to the methyl groups at C-25. The doublets (*J* = 6.4 Hz) at δ 0.94 indicated the methyl group at C-20. Thus, the identity of the compound **5** was confirmed as β-sitosterol by observing similarities in characteristic features on ^1^H NMR spectrum of the compound and those reported for the compound in the table [38]. β-sitosterol is reported for the first time from *Glycosmis cyanocarpa*.

### 2.2. Effect of Isolated Phytochemicals from G. cyanocarpa on DPPH Free Radical Scavenging Activity

The isolated phytochemicals demonstrated a dose-dependent free radical scavenging activity in comparison to the reference in the DPPH free radical scavenging study. At 200 μg/mL, compound **1** demonstrated significant scavenging activity (92.96%) in comparison to the standard BHT (96.48%) (Figure 3). The linear regression equation, which is depicted in Figure 4, was used to obtain the IC_50_ values of BHT and the phytochemicals.

### 2.3. Effect of Isolated Phytochemicals from G. cyanocarpa on Brine Shrimp Lethality Bioassay

The isolated phytochemicals showed dose-dependent mortality when compared to standard in the brine shrimp lethality bioassay (Figure 5 and Figure 6). Compound **2** demonstrated highest cytotoxicity with an LC_50_ value of 1.02 µg/mL compared to the standard with an LC_50_ value of 0.91 µg/mL. Besides, the phytochemicals extracted from *G. cyanocarpa* exhibited LC_50_ values ranging from 1.02 µg/mL (Compound **2**) to 1.92 µg/mL (Compound **4**) (Figure 7).

### 2.4. Effect of Isolated Phytochemicals from G. cyanocarpa on Disc Diffusion Assay

With the antibiotics vancomycin, azithromycin, tetracycline and levofloxacin serving as standards, all of the phytochemicals had their antibacterial properties against two Gram-positive and four Gram-negative bacteria evaluated (Table 1). The range of different phytochemicals’ zones of inhibition was 15 to 21 mm. Compounds **1** and **3** showed broad-spectrum antibacterial activity in a disc diffusion study against both Gram-positive and Gram-negative bacteria, whereas Compounds **2**, **4** and **5** inhibited only Gram-positive bacteria, making them possible sources for antimicrobial agents for such kind of bacteria.

## 3. Discussion

Antioxidants are chemicals that can reduce the production of reactive oxygen species (ROS) [43]. They are essential for the body’s ability to combat oxidative stress and the harmful effects of ROS [44]. In the DPPH free radical scavenging assay, the phenolic compound (**1**) and the indole alkaloid (**2**) demonstrated remarkable antioxidant activity with IC_50_ values of 9.97 and 12.37 μg/mL, respectively, compared to the standard BHT with IC_50_ value 6.06 μg/mL. β-sitosterol (**5**) exhibited moderate antioxidant activity with IC_50_ value 23.65 μg/mL. With more than 8000 structures now known, phenolic compounds, also known as polyphenols, are among the most common and widely dispersed groups of plant secondary metabolites [45]. According to numerous studies, plant phenolics make up the largest group of secondary metabolites that act as primary antioxidants or free radical scavengers. By preserving a balance between oxidants and antioxidants, phenolic compounds help the body fight oxidative stress [46]. A phenolic compound’s antioxidant action may be mediated via a hydrogen atom transfer (HAT), single-electron transfer via proton transfer (SET-PT), sequential proton loss electron transfer, transition metal chelation (TMC), or all of these mechanisms [47]. The position of functional groups around the nuclear structure affects the free radical scavenging and antioxidant properties of phenolic compounds. The primary structural factors affecting the antioxidant ability of phenolics are the number and configuration of H-donating hydroxyl groups. The high antioxidant activity of compound **1** may be attributable to its hydroxyl groups that are potent hydrogen doners. Additionally, by allowing electron delocalization across the molecule, the conjugated double bonds may stabilize the phenoxy radical [45,48]. Indoles are a common component of a wide range of bioactive alkaloids, agrochemicals and pharmaceuticals. The indole ring system is an important structural moiety with a variety of pharmacological properties, including antioxidant, antifungal, antibacterial, antihistaminic, anti-HIV, anticonvulsant, plant growth regulator, analgesic and anti-inflammatory [49]. Indoles are well known for their amazing antioxidant properties, which prevent biological systems from peroxidation of both proteins and lipids [50]. The heterocyclic nitrogen atom can be considered as an active redox center of compound **2** due to the presence of a free electron pair there. Therefore, the relocation of this electron pair in the aromatic system may appear to be crucial for the antioxidant activity of indole derivatives. The SET-PT mechanism might involve the transfer of one electron from the nitrogen atom, which may result in the formation of a cation radical. Compound **2** may also transfer a hydrogen atom (represented by the N-H group) to the DPPH radical, resulting in the formation of a resonance-stabilized indolyl radical [51]. Compound **5** might exhibit scavenging activity by showing protection against depletion of antioxidants such as catalase, superoxide dismutase, glutathione peroxidase, glutathione reductase, glutathione S-transferase and increasing nonenzymatic antioxidant levels (vitamin C, vitamin E and glutathione) as these are the antioxidant mechanisms associated with phytosterols [52]. To fully understand the fundamental antioxidant mechanism of our separated phytochemicals, more research is required.

In the brine shrimp lethality bioassay, the indole alkaloid (**2**) and the phenolic compound (**1**) exhibited outstanding cytotoxic activity with LC_50_ values of 1.02 and 1.08 μg/mL, respectively, compared to the standard VS with LC_50_ value 0.91 μg/mL. The polyprenol (**3**) and β-sitosterol (**5**) demonstrated moderate cytotoxicity with LC_50_ values of 1.39 and 1.37 μg/mL. Many indole alkaloids and their derivatives, namely Vinblastine, Vincristine, Vinorelbine, Rucaparib camsylate, Alectinib hydrochloride, Osimertinib, Anlotinib Dihydrochloride, Panobinostat, etc., are commonly applied in clinical practice to treat various types of cancer, such as acute leukemia, malignant lymphoma, small-cell lung cancer and breast cancer [27]. According to a mechanistic research, indole alkaloids participate in the PI3K/Akt/mTOR signaling route, the MAPK signaling system, the ROS signaling pathway and the Beclin-1 signaling pathway to regulate autophagy, which results in anticancer activity [27]. Inhibition of COX-2 enzyme can also be attributed to the cytotoxic potential of compound **2** [53]. Previously studied mechanisms of indole alkaloids include tubulin polymerization, inhibition of histone deacetylases (HDACs), sirtuins, PIM kinases, DNA topoisomerases and σ receptors [54,55]. Studies on the structure activity relationship have highlighted the importance of aromatic rings and hydroxylic groups in demonstrating powerful anti-cancer properties of phenolic compounds [56]. Numerous mechanisms, including cell proliferation, invasion, autophagy and apoptosis may be linked to the anticancer activity of the compound **1**. The previously published literature revealed the cytotoxicity mechanism of different phenolic compounds, for example, downregulating the expression of PI3K/Akt and decreasing Bcl2/Bax (anthocyanin), promoting autophagy through the IRE1-JNK-CHOP pathway in gastric cancer cells while suppressing the production of vascular endothelial growth factor (kaempferol), inhibiting the G0/G1 phase In HeLa and CaSki cervical cancer cells (ferulic acid), inducing the expression of p53 and p21 and suppresses the production of cyclin D1 and cyclin E (ferulic acid), stimulating antitelomerase activity (gallic acid), activating AMPK and reducing the IL-6 expression gene (ellagic acid) [57]. Compounds **3** and **5** might show anticancer effect through apoptosis and cell cycle arrest [58,59]. Additional research is necessary to comprehend the fundamental cytotoxicity mechanism of our isolated phytochemicals.

Newer antibacterial agents can act as a handy tool to combat bacterial infections [60,61]. Compounds **1** and **3** demonstrated broad spectrum antibacterial activity against both Gram-positive and Gram-negative bacteria in disc diffusion study. According to certain research, the quantity and positioning of hydroxyl groups may have a considerable impact on the antibacterial activity of compound **1**. Due to the presence of hydroxyl groups, it is commonly accepted that phenolic compounds work primarily against the cytoplasmic membrane of bacterial cells to exert their antibacterial effects. The buildup of hydrophobic phenolic groups in the lipid bilayer may interfere with lipid–protein interactions and enhance membrane permeability, leading to further changes in membrane structure and speeding up the extensive leakage of intracellular components until membrane integrity is completely destroyed to allow the entry of more antibacterial agents [62]. Due to the extra protection the outer membrane provides, Gram-negative bacteria are typically more resistant to antimicrobial treatments than Gram-positive bacteria. The presence of porin proteins, which make it easy for tiny molecules with masses under 500 Da to get through, makes the outer membrane more permeable than the cell membrane. However, loss of Mg^2+^ results in loosening or disruption to the outer membrane, which makes it easier for bigger molecules to get through the cell membrane and cause bacterial toxicity. Compound **3** has substantially longer hydrocarbon tails that should make it more effective at increasing membrane permeability [63].

In the same study, compounds **2**, **4** and **5** exhibited inhibitory action only against Gram positive bacteria. The nitrogen atom in the indole ring of compound **2** preserves the aromatic system and causes N-H bond to be acidic rather than basic. The N-H moiety and π–π stacking or cation- π interactions of the aromatic moiety might enable the indole ring to establish hydrogen bonds. As it may bind protein kinase enzymes with good affinity and inhibit autophosphorylation. Given that, we think that the stronger inhibition against Gram-positive bacteria may be predominantly caused by the bacterial penetration [64]. The antibacterial effects of the compounds **4** and **5** might be a result of their capacity to inhibit sortase, an enzyme involved in the secretion and anchoring of cell wall proteins. One potential mechanism for the effects of sterols on bacteria is membrane disruption [65].

## 4. Materials and Methods

### 4.1. Sample Collection and Preparation

Stem and leaf of *Glycosmis cyanocarpa* were harvested from Narshingdi, Bangladesh in July 2019. An expert from the Bangladesh National Herbarium (BNH) correctly identified the plant, and a specimen voucher with the accession number 48257 was submitted for this collection’s future use. The plant components were thoroughly cleaned. The plant components were divided into little pieces and dried in the shade for seven days. The dried material was then carefully ground into a coarse powder using a high-quality grinding apparatus. A total of 1.5 kg of the finished product was sampled.

### 4.2. Instrumentations, Drugs and Chemicals

NMR spectra in deuterated chloroform were captured using a Bruker (400 MHz) instrument (CDCl_3_). Buchi Rotavapor performed the solvent evaporation (Essen, Germany). Vacuum Liquid Chromatography (VLC) and Gel Permeation Chromatography (GPC) were both carried out using Kieselgel 60H and Sephadex LH 20 (Sigma-Aldrich, St. Louis, Missouri, USA), respectively. Precoated thin layer chromatography plates were used for the compound analysis (Silica gel 60 F 254, Merck, Germany). The spots on the TLC plates were seen using UV light and vanillin/H_2_SO_4_ reagents. The remainder of the reagents and solvents were all analytical-grade and bought from a reputable supplier (Active Fine Chemicals Ltd., Bangladesh; Merck, Germany; DaeJung, Korea). The source for vincristine sulfate and butylated hydroxy anisole was Opsonin Pharma Ltd., Dhaka, Bangladesh.

### 4.3. Test Microorganism

The antimicrobial assay used both Gram-positive bacteria (*Sarcina lutea* and *Staphylococcus aureus*) and Gram-negative bacteria (*Salmonella typhi*, *Escherichia coli*, *Salmonella typhi*, *Shigella flexneri* and *Klebsiella* spp.) which was supplied by the University of Dhaka in Bangladesh.

### 4.4. Experimental Design

#### 4.4.1. Extraction of Plant Material

A clean, amber-colored bottle was used to soak 1.5 kg of air-dried and powdered material of *G. cyanocarpa* in 10 litres of distilled methanol (MeOH) for two weeks while sometimes stirring and shaking. After two weeks of cold extraction, the mixture was filtered using a Buchii Rotavapour (Essen, Germany) and a cotton plug in a big funnel, which lowered the amount of filtrate. Over the course of six days, this procedure was repeated several times, and dried extracts were gathered in the same beaker. The final product was 42 g (2.8%) of dried methanol extract.

#### 4.4.2. Isolation of Compounds

A portion of the crude extract (20 g) was subjected to Vacuum Liquid Chromatography (VLC) using hexane, ethyl acetate (EtOAc), and MeOH with increasing polarity [66]. Altogether 40 VLC fractions were collected. All the VLC fractions were fractionated on a Sephadex LH-20 column into subfractions each using CHCl_3_ as the eluting solvent. Table 2 represents the summary of the compounds isolated.

#### 4.4.3. Structural Identification of the Compounds

The ^1^H NMR spectra of compounds **1**–**5** in deuterated chloroform (CDCl_3_) were measured at 400 MHz using a Bruker 400 NMR spectrometer, and the values are presented in relation to the residual non-deuterated solvent signal. The coupling constants are given in Hertz (Hz). Chemical shifts are quantified in δppm.

### 4.5. Antioxidant Assay

#### DPPH Free Radical Scavenging Assay

To evaluate the free radical scavenging abilities of isolated phytochemicals on 1, 1-diphenyl-2-picrylhydrazyl, 2.0 mL of methanol solution of the compounds at serially diluted various concentrations (200 μg/mL to 0.78125 μg/mL) were mixed with 3.0 mL of a DPPH methanol solution (20 μg/mL) (DPPH). The mixture was kept in dark for 1 h. The antioxidant properties were assessed by comparing the decolorizing of a purple DPPH methanol solution by the compounds to that of butylated hydroxy anisole (BHA) [67,68].
(1)$$\%\;Inhibition\;of\;free\;radical\;DPPH=\left(1-\frac{Absorbance\;ofsample}{Absorbance\;of\;the\;control\;reaction}\right)\times 100$$

### 4.6. Cytotoxicity Assay (Brine Shrimp Lethality Bioassay)

The brine shrimp lethality bioassay was conducted to evaluate the cytotoxic potential of the isolated pure compounds. An amount of 38 g of NaCl salt was dissolved in 1000 mL of distilled water to simulate seawater, along with NaOH to keep the pH constant (8.0). Brine shrimp eggs were hatched in artificial seawater to produce nauplii. Isolated compounds were dissolved in DMSO then diluted with simulated seawater into a series of concentrations (400–0.78125 μg/mL). As the reference standard, vincristine sulfate was employed at various doses (400–0.78125 μg/mL), and DMSO served as the negative control. The nauplii were counted visually, and then 5 mL of simulated saltwater was added to vials of the mixture at room temperature [68,69].
(2)$$Mortality\left(\%\right)=\frac{Number\;of\;nauplii\;death}{Number\;of\;nauplii\;taken}\times 100$$

### 4.7. Antibacterial Assay (Disc Diffusion Test)

The disc diffusion method was used to evaluate the antibacterial potential of the isolated phytochemicals [68,70]. The test samples (100 μg/disc) were evenly distributed on 6 mm desiccated, sterilized filter paper discs (Whatman No. 1) on nutrient agar N.A. medium that had already been pre-inoculated with test bacteria. For pre-inoculation, four or five colonies were taken with a wire loop from a pure bacterial culture (not more than 48 h). The colonies were mixed with 0.9% saline solution to achieve bacterial suspension into which a sterile swab was dipped. Then the agar was streaked with the swab for inoculation. Sterilized blank discs were used as the negative standard and four commercially available antibiotic discs (Vancomycin, Azithromycin, Tetracycline and Levofloxacin) as the positive standard (100 μg/disc). To allow for the greatest possible diffusion of test samples into the medium, the plates were kept at a low temperature (4 °C) for around 24 h upside down. The plates were then turned over and kept at 37 °C for 24 h. The antibiotic samples significantly infiltrated the medium and prevented the growth of microorganisms, which was evident as a clear, distinct region designated as a zone of inhibition. To calculate the antibacterial activity of the test samples, the millimeter widths of the zones of inhibition were assessed.

## 5. Conclusions

In this experiment, compounds **1** and **2** demonstrated the highest free radical scavenging and cytotoxic activities, respectively. Again, all the isolated compounds (**1**–**5**) exhibited antimicrobial activity against Gram-positive bacteria and compounds **1** and **2** against both Gram-positive and Gram-negative bacteria. After sequential chromatographic separation and purification, the methanol extract of the stem and leaf of *G. cyanocarpa* produced a total of five phytochemicals including one new natural compound (**1**). It is still advised to do in-depth research on this plant to ascertain their precise pharmacological effects as well as their safety profile and to identify additional bioactive phytochemicals and isolate them.

## Figures and Tables

**Figure 1 molecules-27-05972-f001:**
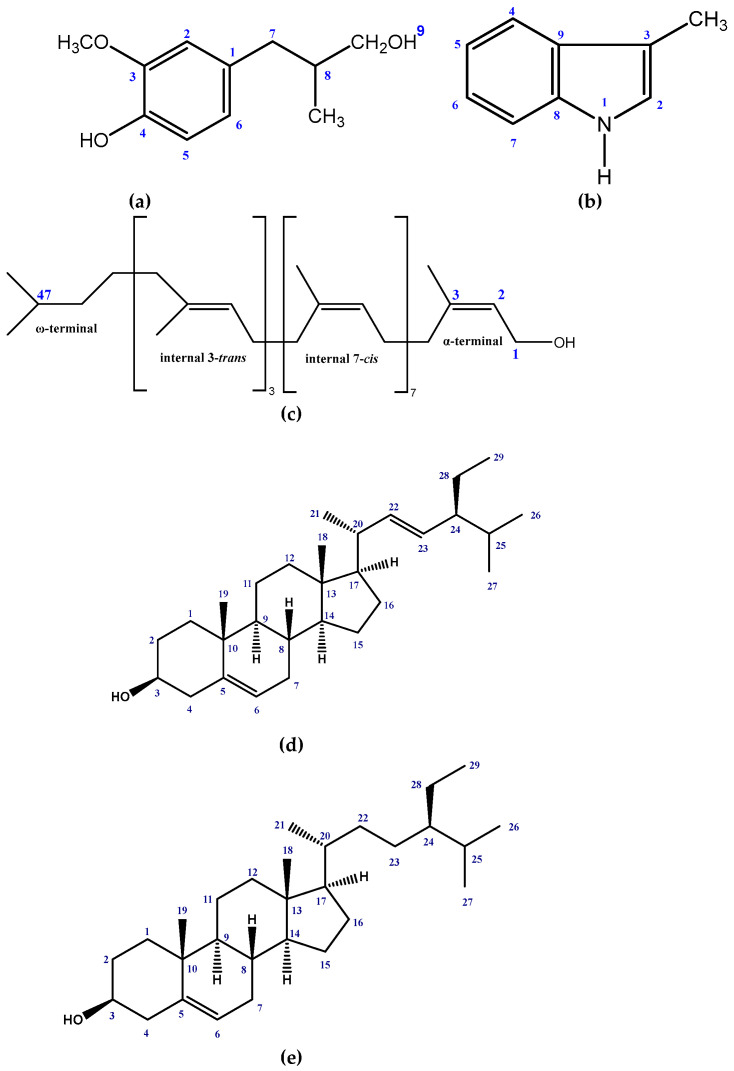
Structures of Isolated phytochemicals from *Glycosmis cyanocarpa* using NMR techniques: (**a**) 4-(3-hydroxy-2-methylpropyl)-2-methoxyphenol, (**b**) 3-methyl-1*H*-Indole, (**c**) Tri-transpoly-cis prenol-12, (**d**) Stigmasterol and (**e**) β-Sitosterol.

**Figure 2 molecules-27-05972-f002:**
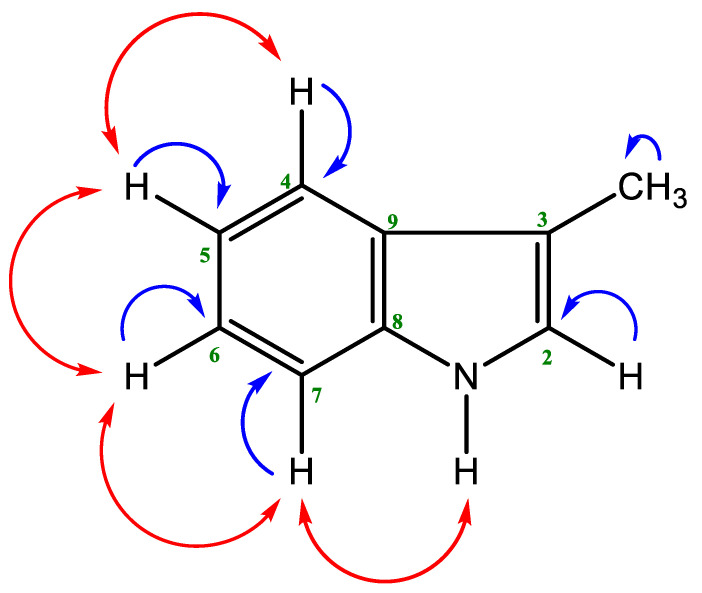
Key HSQC (blue) and COSY (red) correlations for compound **2**.

**Figure 3 molecules-27-05972-f003:**
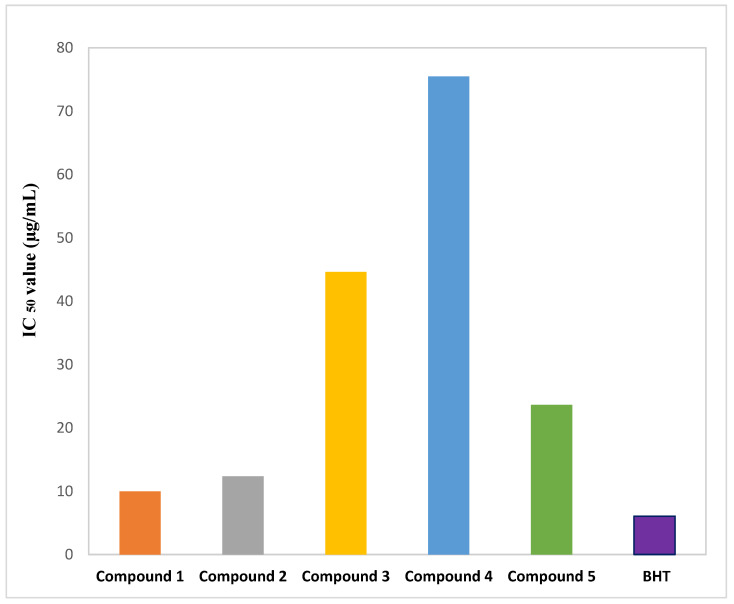
IC_50_ values of Butylated Hydroxytoluene (BHT) and isolated phytochemicals of *Glycosmis cyanocarpa*.

**Figure 4 molecules-27-05972-f004:**
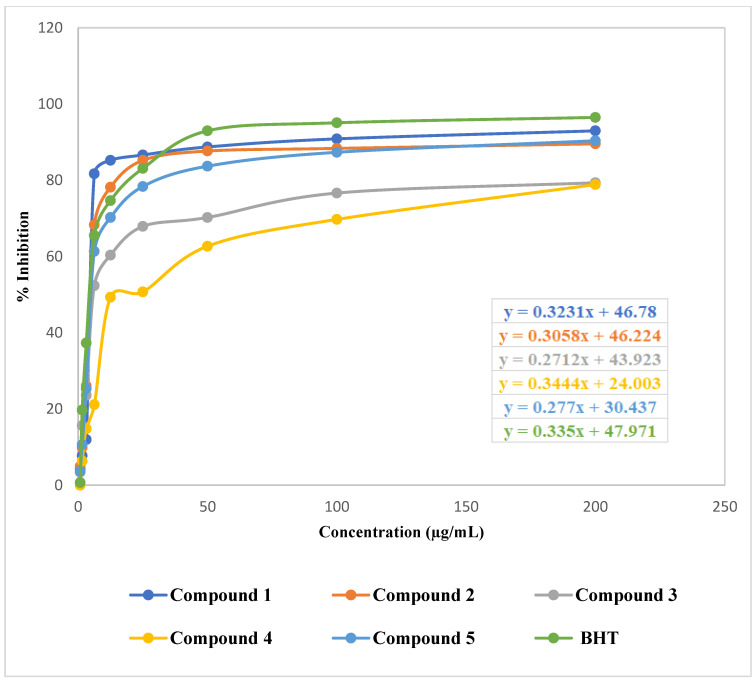
Linear regression equations (IC_50_) of Butylated Hydroxytoluene (BHT) and isolated phytochemicals of *Glycosmis cyanocarpa*.

**Figure 5 molecules-27-05972-f005:**
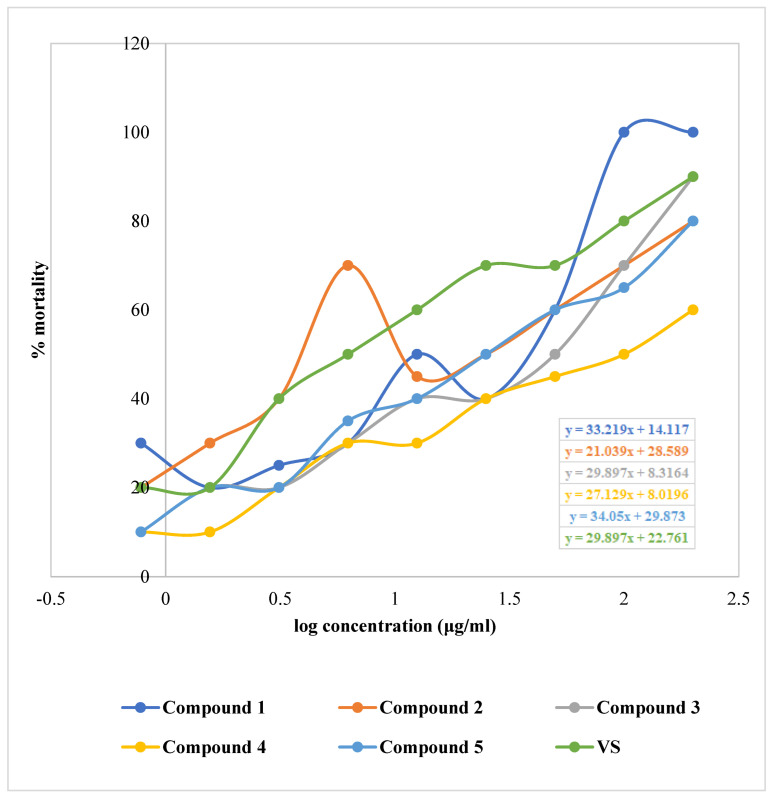
Linear regression equations (LC_50_) of Vincristine Sulphate (VS) and isolated phytochemicals of *Glycosmis cyanocarpa*.

**Figure 6 molecules-27-05972-f006:**
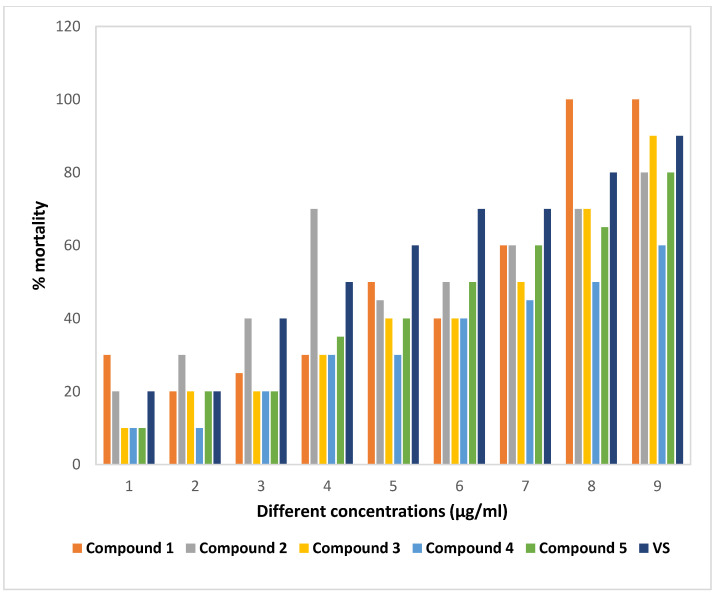
Percentage of mortality in Vincristine Sulphate (VS) and isolated phytochemicals of *Glycosmis cyanocarpa* where log concentrations −0.10721, 0.19382, 0.49485, 0.79588, 1.09691, 1.39794, 1.69897 and 2.30103 μg/mL are denoted serially from 1 to 9 in X-axis.

**Figure 7 molecules-27-05972-f007:**
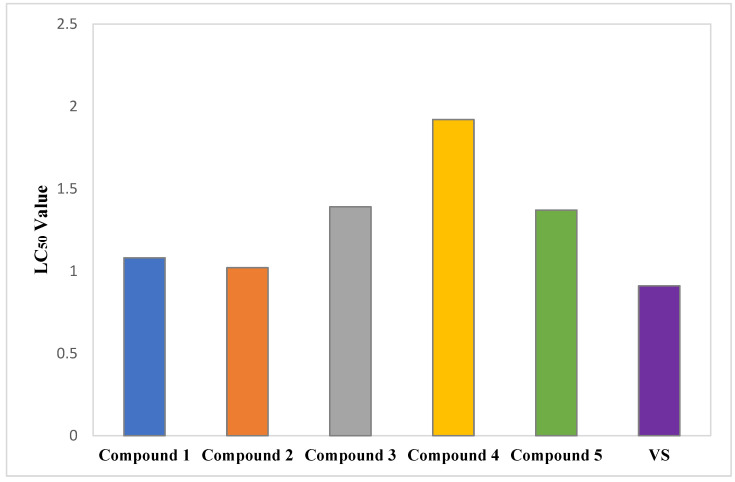
LC_50_ values of Vincristine Sulphate (VS) and isolated phytochemicals of *Glycosmis cyanocarpa*.

**Table 1 molecules-27-05972-t001:** Antibacterial Activity of the Isolated Compounds.

Test Microorganisms	Diameter of Zone of Inhibition (mm)					Vancomycin	Azithromycin	Tetracycline	Levofloxacin
	Compounds								
	1	2	3	4	5				
**Gram-Positive Bacteria**									
*Staphylococcus aureus*	21	17	19	16	17	41	39	37	40
*Sarcina lutea*	20	19	-	17	15	42	40	38	39
**Gram-Negative Bacteria**									
*Salmonella typhi*	18	-	17	-	-	39	34	41	36
*Klebsiella spp.*	-	-	-	-	-	43	37	39	38
*Shigella flexneri*	-	-	-	-	-	38	41	38	41
*Escherichia coli*	19	-	18	-	-	42	40	41	42

**Table 2 molecules-27-05972-t002:** Summary of the isolated compounds from the crude methanol extract of *Glycosmis cyanocarpa*.

Compounds.	VLC Fraction No.	Sephadex Fraction No.	Further Purification Steps	R_f_	UV Visualization	Color upon Spraying Vanillin Sulphate
Compound **1**	15	56–69	PTLC using E: T = 25:75	0.14	Quenching	Yellow, turns black upon heating
Compound **2**	14–15	77–101	PTLC using E: T = 15:85	0.35	Dark Quenching	Orange red, turns purple upon heating
Compound **3**	1–3	3–13	PTLC using E: T = 1:99	0.84	Quenching	Pale yellow
Compound **4**	1–3	26–30	Crystals obtained from concentrated sephadex fraction	0.30	Not observed	Purple, turns black upon heating
Compound **5**	4–9	* 192–231	Solid crystal obtained	0.34	Not observed	Purple, turns black upon heating

* = compound was isolated from the silica column.

## Data Availability

Not applicable.

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
