# Peer review of "Isolation, Characterization and Pharmacological Investigations of a New Phenolic Compound along with Four Others Firstly Reported Phytochemicals from Glycosmis cyanocarpa (Blume) Spreng"

_molecules, 2022, doi:10.3390/molecules27185972_

Round 1

Reviewer 1 Report

The manuscript "Isolation and characterization of a new phenolic compound (4- (3-hydroxy-2-methylpropyl)-2-methoxyphenol) and four other firstly reported phytochemicals from Glycosmis cyanocarpa (Blume) Spreng. along with determination of their pharmacological potentials" is a well-formulated and executed study, but title is very long. However, the concerns below need to be addressed.

The manuscript needs to be grammar and spell-checked. The authors need to follow same format for units and citations throughout the manuscript.

Line 28: Staphylococcus aureus (Italic font).

Line 30: Supplement: what does this word mean here?

Introduction: should be update (use new references)

(Line 193: g/mL, is it in grams or milligrams?!

Line 193: compound 4 is the greatest in the scavenging activity while compound 1 is the smallest! what about the others?

Line 201: Figure 3, Where are the color keys for other compounds?

Line 209: 1.02, What is the measurement unit? is it microgram/ml?

Line 211: Figure 5, I think here it is better to use non linear regression so the curves will be clearer.

Line 214: Figure 6 Concentration in chart, what does that mean?

Line 223: agents, Add "for such kind of bacteria", please.

Line 225: in Table 1: Should involve the control antibiotics also, Italic font for all bacterial strains used here

Line 346: 1.5 kg of the finished product was sampled, How much row material did you used? to calculate the yield

Line 366: methanol, hexane and ethyl acetate, which company?

Line 398: Rephrase the sentence in the correct context, please. it should be: plant extracts were dissolved in DMSO then diluted with simulated seawater into a series of concentrations (400-0.78125 mg/ml).

Line 404: Mention as "Antibacteria Assay" please.

Line 406: Are they strile? if yes, then in which way did were they sterilized? Which concentrations were used for the samples?

Line 407: Which type and which No. of filter paper?

Line 407: Its officially known as nutrient agar N. A. medium.

Line 408: pre-inoculated, Mention the technique of inoculation and the bacterial concentration used for incubation, please.

Line 408: Blank discs, Mention if they are strile also, please.

Line 409: antibiotic discs, Mention their concentration in each disk, please.

Line 412: upside down, How did the researcher make sure that the disks will not fall down when flipping the plates upside down?!

Usually when antibacterial assay is performed, plates would be incubated in their normal way (not upside down) whether if it is in the refreg. or in the incubator and whether it was performed by disk diffusion assay or well diffusion assay.

Line 412 4oC, Degree should not be written like this.

Author Response

At first we are thanking the reviewer for their valuable suggestions to improve the overall quality of the existing manuscript. We have revised the manuscript point-by-point as per the suggestion of the reviewers. Please find the   Authors’ response below:

Reviewer 1

  1. The manuscript needs to be grammar and spell-checked. The authors need to follow same format for units and citations throughout the manuscript.

Authors’ Reply: The whole manuscript has been checked and corrected where necessary.

  1. Line 28: Staphylococcus aureus (Italic font).

Authors’ Reply: It is corrected as per the suggestion.

  1. Line 30: Supplement: what does this word mean here?-

Authors’ Reply: It was mistakenly written here and now removed as per the suggestion.

  1. Introduction: should be update (use new references)- Updated with new references

Authors’ Reply: We have rejuvenated the introduction section with new references as per the suggestion.

  1. Line 193: g/mL, is it in grams or milligrams?!

Authors’ Reply:  It will be micrograms and corrected accordingly.

  1. Line 193: compound 4 is the greatest in the scavenging activity while compound 1 is the smallest! what about the others?

Authors’ Reply: In the DPPH free radical scavenging assay, the phenolic compound (1) and the indole alkaloid (2) demonstrated remarkable antioxidant activity with IC50 values of 9.97 and 12.37 μg/mL, respectively, compared to the standard BHT with an IC50 value 6.06 μg/mL. β-sitosterol (5) exhibited moderate antioxidant activity with an IC50 value 23.65 μg/mL. Compounds 3 & 4 showed moderate activity. The topic has been discussed elaborately in the discussion chapter and also depicted in figures 3, 4, and 5.  We have mentioned only the best value here to avoid repetition of information (In Figure 2 and Discussion section).

  1. Line 201: Figure 3, Where are the color keys for other compounds?

Authors’ Reply: It is corrected as per the suggestion.

  1. Line 209: 1.02, What is the measurement unit? is it microgram/ml?

Authors’ Reply: It is µg/ml and corrected accordingly.

  1. Line 211: Figure 5, I think here it is better to use non linear regression so the curves will be clearer.

Authors’ Reply: It is adjusted as per the suggestion of respected reviewer.

  1. Line 214: Figure 6 Concentration in chart, what does that mean?-

Authors’ Reply: It indicates the concentration of each sample we used during the test. This chart is depicting correlation between the concentration and scavenging activity of each sample.

  1. Line 223: agents, Add "for such kind of bacteria", please.

Authors’ Reply: It is corrected as per the scholarly suggestion of the reviewer.

  1. Line 225: in Table 1: Should involve the control antibiotics also, Italic font for all bacterial strains used here-

Authors’ Reply: It is corrected as per the scholarly suggestion of the reviewer.

  1. Line 346: 1.5 kg of the finished product was sampled, How much row material did you used? to calculate the yield-

Authors’ Reply: We have used 3kg of raw materials.

  1. Line 366: methanol, hexane and ethyl acetate, which company?-

Authors’ Reply: Active Fine Chemicals Ltd, Bangladesh; Merck, Germany; DaeJung, Korea. The information is also added in subchapter 4.2. Instrumentations, Drugs, and Chemicals

  1. Line 398: Rephrase the sentence in the correct context, please. it should be: plant extracts were dissolved in DMSO then diluted with simulated seawater into a series of concentrations (400-0.78125 mg/ml).-

Authors’ Reply: It is corrected as per the scholarly suggestion of the reviewer.

  1. Line 404: Mention as "Antibacteria Assay" please.-

Authors’ Reply: It is corrected as per the scholarly suggestion of the reviewer.

  1. Line 406: Are they strile? if yes, then in which way did were they sterilized? Which concentrations were used for the samples? –

Authors’ Reply: We are thanking reviewer for his scholarly suggestion. The filter paper discs were sterilized by ten minutes exposure on IR lamp. Petri dishes and other glasswares were sterilized by autoclaving at a temperature of 121oC and a pressure of 15-lbs/sq. inch for 20 minutes. The sample concentration was 400μg/disc. Concentration added in manuscript.

  1. Line 407: Which type and which No. of filter paper?-

Authors’ Reply: We have used Whatman No.1 filter paper and the information is added in manuscript

  1. Line 407: Its officially known as nutrient agar N. A. medium.-

Authors’ Reply: It is corrected as per the scholarly suggestion of the reviewer.

  1. Line 408: pre-inoculated, Mention the technique of inoculation and the bacterial concentration used for incubation, please. –

Authors’ Reply: It is corrected as per the scholarly suggestion of the reviewer.

  1. Line 408: Blank discs, Mention if they are strile also, please.-

Authors’ Reply: It is mentioned as per the suggestion of reviewer.

  1. Line 409: antibiotic discs, Mention their concentration in each disk, please.-

Authors’ Reply: It is mentioned as per the suggestion of reviewer.

  1. Line 412: upside down, How did the researcher make sure that the disks will not fall down when flipping the plates upside down?!- The disks were pressed down onto the agar so that it is secure and does not fall off. Usually when antibacterial assay is performed, plates would be incubated in their normal way (not upside down) whether if it is in the refreg. or in the incubator and whether it was performed by disk diffusion assay or well diffusion assay.

Authors’ Reply: We are thanking reviewer for his scholarly comment. Agar plates is generally incubated with the agar face down. The other way (lids on top) can result in water evaporation (mainly from the agar) condensing on the lids and then drops of water falling onto the agar and individual colonies. This will cause smearing and a combination of colonies and/or possible cross-contamination between colonies of different species if present. (Ref- Hudzicki, J., 2009. Kirby-Bauer disk diffusion susceptibility test protocol. American society for microbiology, 15, pp.55-63.)

  1. Line 412 4oC, Degree should not be written like this.

Authors’ Reply: It is corrected as per the scholarly suggestion of the reviewer.

Reviewer 2 Report

General comments:

All the part describing the NMR spectra is so long. Is all the information there necessary?

The discussion section is too much focused on the literature and the results obtained by other researchers. The authors are advised to focus in interpretation of their own results and discuss them properly.

Some references, and not few, are very old. It is recommended that references not older than 10 years to be used, except the cases when a reference is essential.  

Concrete comments:

Title – please write a more representative and if possible a shorter title “Isolation and characterization of a new phenolic compound….. and four other” sounds confusing

Line 19 – “The chemical studies led to the isolation” please rewrite that in a more concrete and scientific way

Line 30 – “respectivelySupplement” – something is wrong here

Line 39 – the authors wrote “have no direct role in plant development and reproduction”. Please mention the role of secondary metabolites (i.e protection against unfavourable environmental conditions (salinity, drought, etc), etc).

Lines 40-41 –  The authors wrote: “These chemicals have pharmacological effects on the human body. The most potent biologically active substances are flavonoids, alkaloids, phenolic compounds, and tannins [1].”

Comment: not all secondary metabolites have pharmacological effects, and it was not determinate that the mentioned categories are the most potent, although I agree that they are the most known. Please re-write that part, add appropriate references, if possible, for each category, or a review article presenting the topic. The reference mentioned (ref 1) has nothing in common with the information stated. Also, please add cyclitols and maybe saponins to the list of mentioned secondary metabolites (as ref for cyclitols can be used DOI: 10.1002/jssc.201900539, or any other)

Lines 42-43 – The authors wrote “Therefore, the separation and purification of secondary metabolites from plants are increasingly getting popular”. Instead of this sentence I would write that something concrete and scientifically, for example “There are a lot of methods and techniques used for extraction, separation and purification of secondary metabolites” and then, I would reference a review article describing such techniques (for example DOI: 10.1002/elps.201700431).

Line 53 – “and other substances” please change that or delete it. We cannot call lipids, blood vessel walls, carbohydrates and DNA “substances”. Moreover, please be carefully with this statement I doubt that ROS are affecting the DNA. Also, I do not know why carbohydrates are mentioned here. To the best of my knowledge, carbohydrates, are sugar molecules, one of three main nutrients found in foods and drinks.

Lines 54-68 Please delete all the paragraph “In terms of severity, after cardiovascular disorder ……and fewer side effect”. It has nothing in common with the manuscript.

Lines 69-70  - The authors wrote: “Here, herbal medication is a very viable solution” – please make it softer, because the topic is debatable.

Lines 70-71  - Please delete the next sentence “For the conditions listed above, there are numerous therapy options”

Figure 1 – it takes more than a page, is it possible to arrange it in a single smaller and nice figure? Not that is something wrong in keeping it as it is, but is not esthetical

Lines 89-119 – the whole text is difficult to follow. Is it possible to convert it to a Table?

Line 143 - In line 133, the authors wrote “The indole and 2-quinolone alkaloids……. ” than, in line 143 they mentioned about 3-methyl-1H-Indole. Question: is 3-methyl-1H-Indole an alkaloid? It knew that it is produced during the anoxic metabolism of L-tryptophan in the mammalian digestive tract, and that it is also found as a natural product in Tachigali glauca, Coprinopsis picacea, and other organisms.

Line 198 - Figure 2, from page 6, should be Figure 3, and then, the number of each Figure should be increased with one.

Line 194 – the authors wrote “The linear regression equation, which is depicted in Figures 4 & 5,…” I think the correct numbers are 4 and 6, but however is a total mess with the figures. Please put correct numbers in the text and Figures captions. Moreover, from the legend of “Figure 3” Compound 4, Compound 5 and BHT are missed.

Lines 227-234 – this paragraph is more suitable for the introduction, can be paste there to replace the part described in lines 54-68

Line 275- please delete the first sentence. The authors did not investigate the effect of the 5 isolated targets against cancer, so they cannot discuss that by some extension and possible connections.

Lines 280-283 – the authors wrote: “Many indole alkaloids and their derivatives namely Vinblastine, Vincristine, Vinorelbine, Rucaparib camsylate, Alectinib hydrochloride, Osimertinib, Anlotinib Dihydrochloride, Panobinostat etc are commonly applied in clinical practice to treat various types of cancer, such as acute leukemia, malignant lymphoma, small-cell lung cancer, and breast cancer”. Please delete that sentence or add a relevant reference that support your statement.

Lines 288-290 the authors wrote: “Recently studied mechanisms….. [36.37]”. I wouldn’t say that an article published 15 years before in a recent study (ref 37).

Lines 306-308 – please delete that sentence. It is not relevant for your article, and nobody can guarantee that bacteria will not develop resistance to natural antimicrobial medications.

Lines 365-366 – in which volume of methanol was the dried material macerated?

Lines 388-393 – after mixing the extract with DPPH, was the mixture kept in the dark before measuring the absorbance? If yes, for how long time? Please include this information into the manuscript.

Author Response

At first we are thanking reviewer for their valuable suggestions to improve the overall quality of the existing manuscript. We have revised the manuscript point-by-point as per the suggestion of the reviewers. Please find the   Authors’ response below:

Reviewer 2

  1. All the part describing the NMR spectra is so long. Is all the information there necessary?

Authors’ Reply: We are thanking reviewer for his scholarly comment. The NMR spectra are described elaborately to strengthen the establishment of the structures of the isolated phytochemicals. To avoid clumsiness, we have presented the NMR spectra and tables in supplementary file.

  1. The discussion section is too much focused on the literature and the results obtained by other researchers. The authors are advised to focus in interpretation of their own results and discuss them properly.

Authors’ Reply: We are thanking reviewer for his scholarly comment. The discussion session has been arranged to support and justify the biological activities of the isolated phytochemicals. An extensive literature study was performed to correlate the activities given by the phytochemicals to their underlying mechanism of action. However, we have rearranged the discussion section to make it more appropriate.

  1. Some references, and not few, are very old. It is recommended that references not older than 10 years to be used, except the cases when a reference is essential.  

Authors’ Reply: Some of the references have been updated. But few remaining old references are essential to establish and validate the chemical structures of the isolated compounds and any analytical procedures behind them. 

  1. Title – please write a more representative and if possible a shorter title “Isolation and characterization of a new phenolic compound….. and four other” sounds confusing Authors’ Reply: It is corrected as per the scholarly suggestion of the reviewer.

  1. Line 19 – “The chemical studies led to the isolation” please rewrite that in a more concrete and scientific way

Authors’ Reply: It is corrected as per the scholarly suggestion of the reviewer.

  1. Line 30 – “respectivelySupplement” – something is wrong here-

Authors’ Reply: It is corrected as per the scholarly suggestion of the reviewer.

  1. Line 39 – the authors wrote “have no direct role in plant development and reproduction”. Please mention the role of secondary metabolites (i.e protection against unfavorable environmental conditions (salinity, drought, etc), etc). –

Authors’ Reply: It is corrected as per the scholarly suggestion of the reviewer.

  1. Lines 40-41 – The authors wrote: “These chemicals have pharmacological effects on the human body. The most potent biologically active substances are flavonoids, alkaloids, phenolic compounds, and tannins [1].”-

Authors’ Reply: It is corrected as per the scholarly suggestion of the reviewer.

  1. Comment: not all secondary metabolites have pharmacological effects, and it was not determinate that the mentioned categories are the most potent, although I agree that they are the most known. Please re-write that part, add appropriate references, if possible, for each category, or a review article presenting the topic. The reference mentioned (ref 1) has nothing in common with the information stated. Also, please add cyclitols and maybe saponins to the list of mentioned secondary metabolites (as ref for cyclitols can be used DOI: 10.1002/jssc.201900539, or any other)

Authors’ Reply: We are thanking reviewer for his valuable suggestion to improve the overall quality of the manuscript. The raised point is corrected as per the scholarly suggestion of the reviewer.

  1. Lines 42-43 – The authors wrote “Therefore, the separation and purification of secondary metabolites from plants are increasingly getting popular”. Instead of this sentence I would write that something concrete and scientifically, for example “There are a lot of methods and techniques used for extraction, separation and purification of secondary metabolites” and then, I would reference a review article describing such techniques (for example DOI: 10.1002/elps.201700431).-

Authors’ Reply: We are thanking reviewer for his valuable suggestion to improve the overall quality of the manuscript. The raised point is corrected as per the scholarly suggestion of the reviewer.

  1. Line 53 – “and other substances” please change that or delete it. We cannot call lipids, blood vessel walls, carbohydrates and DNA “substances”. Moreover, please be carefully with this statement I doubt that ROS are affecting the DNA. Also, I do not know why carbohydrates are mentioned here. To the best of my knowledge, carbohydrates, are sugar molecules, one of three main nutrients found in foods and drinks.-

Authors’ Reply: We are thanking reviewer for his valuable suggestion to improve the overall quality of the manuscript. The raised point is corrected as per the scholarly suggestion of the reviewer. We have also added a new reference to support the effect of ROS on carbohydrates.

  1. Lines 54-68 Please delete all the paragraph “In terms of severity, after cardiovascular disorder ……and fewer side effect”. It has nothing in common with the manuscript.- Authors’ Reply: We are thanking reviewer for his scholarly opinion. But we are requesting the reviewer to allow us to keep the paragraph as the paragraph is written as a background study to justify the reasons for choosing the performed biological investigations i.e. antimicrobial and anticancer potentials.

  1. Lines 69-70  - The authors wrote: “Here, herbal medication is a very viable solution” – please make it softer, because the topic is debatable.’-

Authors’ Reply: It is corrected as per the scholarly suggestion of the reviewer.

  1. Lines 70-71  - Please delete the next sentence “For the conditions listed above, there are numerous therapy options”-

Authors’ Reply: It is deleted as per the scholarly suggestion of the reviewer.

  1. Figure 1 – it takes more than a page, is it possible to arrange it in a single smaller and nice figure? Not that is something wrong in keeping it as it is, but is not esthetical.

Authors’ Reply: It is rearranged as per the scholarly suggestion of the reviewer.

  1. Lines 89-119 – the whole text is difficult to follow. Is it possible to convert it to a Table?- Authors’ Reply: We are thanking reviewer for his scholarly opinion. The tables for NMR data of each phytochemical have been provided in the supplementary file.

  1. Line 143 - In line 133, the authors wrote “The indole and 2-quinolone alkaloids……. ” than, in line 143 they mentioned about 3-methyl-1H-Indole. Question: is 3-methyl-1H-Indole an alkaloid? It knew that it is produced during the anoxic metabolism of L-tryptophan in the mammalian digestive tract, and that it is also found as a natural product in Tachigali glauca, Coprinopsis picacea, and other organisms.-

Authors’ Reply: We are thanking reviewer for his scholarly opinion. The mentioned phytochemical is a naturally occurring indole class of alkaloid.

  1. Line 198 - Figure 2, from page 6, should be Figure 3, and then, the number of each Figure should be increased with one.-

Authors’ Reply: It is rearranged as per the scholarly suggestion of the reviewer.

  1. Line 194 – the authors wrote “The linear regression equation, which is depicted in Figures 4 & 5,…” I think the correct numbers are 4 and 6, but however is a total mess with the figures. Please put correct numbers in the text and Figures captions. Moreover, from the legend of “Figure 3” Compound 4, Compound 5 and BHT are missed-

Authors’ Reply: It is rejuvenated as per the scholarly suggestion of the reviewer.

  1. Lines 227-234 – this paragraph is more suitable for the introduction, can be paste there to replace the part described in lines 54-68

Authors’ Reply: It is rearranged as per the scholarly suggestion of the reviewer.

  1. Line 275- please delete the first sentence. The authors did not investigate the effect of the 5 isolated targets against cancer, so they cannot discuss that by some extension and possible connections.-

Authors’ Reply: It is deleted as per the scholarly suggestion of the reviewer.

  1. Lines 280-283 – the authors wrote: “Many indole alkaloids and their derivatives namely Vinblastine, Vincristine, Vinorelbine, Rucaparib camsylate, Alectinib hydrochloride, Osimertinib, Anlotinib Dihydrochloride, Panobinostat etc are commonly applied in clinical practice to treat various types of cancer, such as acute leukemia, malignant lymphoma, small-cell lung cancer, and breast cancer”. Please delete that sentence or add a relevant reference that support your statement.-

Authors’ Reply: Reference is added as per the scholarly suggestion of the reviewer.

  1. Lines 288-290 the authors wrote: “Recently studied mechanisms….. [36.37]”. I wouldn’t say that an article published 15 years before in a recent study (ref 37).-

Authors’ Reply: We are thanking author for his scholarly comment. The raised issue is resolved in the revised manuscript.

  1. Lines 306-308 – please delete that sentence. It is not relevant for your article, and nobody can guarantee that bacteria will not develop resistance to natural antimicrobial medications.

Authors’ Reply: We are thanking author for his scholarly comment. The raised issue is resolved in the revised manuscript.

  1. Lines 365-366 – in which volume of methanol was the dried material macerated?- Authors’ Reply: We are thanking author for his scholarly comment. The raised issue is resolved in the revised manuscript.

  1. Lines 388-393 – after mixing the extract with DPPH, was the mixture kept in the dark before measuring the absorbance? If yes, for how long time? Please include this information into the manuscript.-

Authors’ Reply: Authors’ Reply: We are thanking author for his scholarly comment. The raised issue is resolved in the revised manuscript.

Round 2

Reviewer 1 Report

  1. Line 346: 1.5 kg of the finished product was sampled, How much row material did you used? to calculate the yield-

Authors’ Reply: We have used 3kg of raw materials.

you should added in manuscript

Reviewer 2 Report

The authors improved the manuscript and answered to most of the comments, or argued acceptable those points that were not changed. I suggest that the manuscript can be accepted for publication.